# Multi-source Domain Adaptation for Semantic Segmentation

**Sicheng Zhao**[1*†], **Bo Li**[23†], **Xiangyu Yue**[1†], **Yang Gu**[2],
**Pengfei Xu**[2], **Runbo Hu**[2], **Hua Chai**[2], **Kurt Keutzer**[1]
[1]University of California, Berkeley, USA  [2]Didi Chuxing, China
[3]Harbin Institute of Technology, China
schzhao@gmail.com, drluodian@gmail.com, {xyyue,keutzer}@berkeley.edu
{guyangdavid,xupengfeipf,hurunbo,chaihua}@didiglobal.com

## Abstract

Simulation-to-real domain adaptation for semantic segmentation has been actively studied for various applications such as autonomous driving. Existing methods mainly focus on a single-source setting, which cannot easily handle a more practical scenario of multiple sources with different distributions. In this paper, we propose to investigate multi-source domain adaptation for semantic segmentation. Specifically, we design a novel framework, termed Multi-source Adversarial Domain Aggregation Network (MADAN), which can be trained in an end-to-end manner. First, we generate an adapted domain for each source with *dynamic semantic consistency* while aligning at the pixel-level cycle-consistently towards the target. Second, we propose *sub-domain aggregation discriminator* and *cross-domain cycle discriminator* to make different adapted domains more closely aggregated. Finally, feature-level alignment is performed between the aggregated domain and target domain while training the segmentation network. Extensive experiments from synthetic GTA and SYNTHIA to real Cityscapes and BDDS datasets demonstrate that the proposed MADAN model outperforms state-of-the-art approaches. Our source code is released at: `https://github.com/Luodian/MADAN`.

## 1  Introduction

Semantic segmentation assigns a semantic label (*e.g.* car, cyclist, pedestrian, road) to each pixel in an image. This computer vision kernel plays a crucial role in many applications, ranging from autonomous driving [1] and robotic control [2] to medical imaging [3] and fashion recommendation [4]. With the advent of deep learning, especially convolutional neural networks (CNNs), several end-to-end approaches have been proposed for semantic segmentation [5, 6, 7, 8, 9, 10, 11, 12, 13, 14]. Although these methods have achieved promising results, they suffer from some limitations. On the one hand, training these methods requires large-scale labeled data with pixel-level annotations, which is prohibitively expensive and time-consuming to obtain. For example, it takes about 90 minutes to label each image in the Cityscapes dataset [15]. On the other hand, they cannot well generalize their learned knowledge to new domains, because of the presence of *domain shift* or *dataset bias* [16, 17].

To sidestep the cost of data collection and annotation, unlimited amounts of synthetic labeled data can be created from simulators like CARLA and GTA-V [18, 19, 20], thanks to the progress in graphics and simulation infrastructure. To mitigate the gap between different domains, domain adaptation (DA) or knowledge transfer techniques have been proposed [21] with both theoretical analysis [22,

---

23, 24, 25] and algorithm design [26, 27, 28, 29, 30, 31, 32]. Besides the traditional task loss on the labeled source domain, deep unsupervised domain adaptation (UDA) methods are generally trained with another loss to deal with domain shift, such as a discrepancy loss [31, 33, 34, 35], adversarial loss [36, 37, 38, 39, 37, 40, 41, 42, 43, 32, 44], reconstruction loss [30, 45, 46], *etc.* Current simulation-to-real DA methods for semantic segmentation [47, 48, 49, 50, 51, 52, 32, 53, 54, 55, 56] all focus on the single-source setting and do not consider a more practical scenario where the labeled data are collected from multiple sources with different distributions. Simply combining different sources into one source and directly employing single-source DA may not perform well, since images from different source domains may interfere with each other during the learning process [57].

Early efforts on multi-source DA (MDA) used shallow models [58, 59, 60, 61, 62, 63, 64, 65, 66, 67]. Recently, some multi-source deep UDA methods have been proposed which only focus on image classification [68, 69, 70]. Directly extending these MDA methods from classification to segmentation may not perform well due to the following reasons. (1) Segmentation is a structured prediction task, the decision function of which is more involved than classification because it has to resolve the predictions in an exponentially large label space [48, 71]. (2) Current MDA methods mainly focus on feature-level alignment, which only aligns high-level information. This may be enough for coarse-grained classification tasks, but it is obviously insufficient for fine-grained semantic segmentation, which performs pixel-wise prediction. (3) These MDA methods only align each source and target pair. Although different sources are matched towards the target, there may exist significant mis-alignment across different sources.

To address the above challenges, in this paper we propose a novel framework, termed Multi-source Adversarial Domain Aggregation Network (MADAN), which consists of Dynamic Adversarial Image Generation, Adversarial Domain Aggregation, and Feature-aligned Semantic Segmentation. First, for each source, we generate an adapted domain using a Generative Adversarial Network (GAN) [36] with cycle-consistency loss [39], which enforces pixel-level alignment between source images and target images. To preserve the semantics before and after image translation, we propose a novel semantic consistency loss by minimizing the KL divergence between the source predictions of a pretrained segmentation model and the adapted predictions of a *dynamic segmentation model*. Second, instead of training a classifier for each source domain [68, 70], we propose *sub-domain aggregation discriminator* to directly make different adapted domains indistinguishable, and *cross-domain cycle discriminator* to discriminate between the images from each source and the images transferred from other sources. In this way, different adapted domains can be better aggregated into a more unified domain. Finally, the segmentation model is trained based on the aggregated domain, while enforcing feature-level alignment between the aggregated domain and the target domain.

In summary, our contributions are three-fold. (1) We propose to perform domain adaptation for semantic segmentation from multiple sources. To the best of our knowledge, this is the first work on multi-source structured domain adaptation. (2) We design a novel framework termed MADAN to do MDA for semantic segmentation. Besides feature-level alignment, pixel-level alignment is further considered by generating an adapted domain for each source cycle-consistently with a novel dynamic semantic consistency loss. Sub-domain aggregation discriminator and cross-domain cycle discriminator are proposed to better align different adapted domains. (3) We conduct extensive experiments from synthetic GTA [18] and SYNTHIA [19] to real Cityscapes [15] and BDDS [72] datasets, and the results demonstrate the effectiveness of our proposed MADAN model.

## 2 Problem Setup

We consider the unsupervised MDA scenario, in which there are multiple labeled source domains $S_1, S_2, \cdots, S_M$, where $M$ is number of sources, and one unlabeled target domain $T$. In the $i$th source domain $S_i$, suppose $X_i = \{\mathbf{x}_i^j\}_{j=1}^{N_i}$ and $Y_i = \{\mathbf{y}_i^j\}_{j=1}^{N_i}$ are the observed data and corresponding labels drawn from the source distribution $p_i(\mathbf{x}, \mathbf{y})$, where $N_i$ is the number of samples in $S_i$. In the target domain $T$, let $X_T = \{\mathbf{x}_T^j\}_{j=1}^{N_T}$ denote the target data drawn from the target distribution $p_T(\mathbf{x}, \mathbf{y})$ without label observation, where $N_T$ is the number of target samples. Unless otherwise specified, we have two assumptions: (1) homogeneity, *i.e.* $\mathbf{x}_i^j \in \mathbb{R}^d, \mathbf{x}_T^j \in \mathbb{R}^d$, indicating that the data from different domains are observed in the same image space but with different distributions; (2) closed set, *i.e.* $\mathbf{y}_i^j \in \mathcal{Y}, \mathbf{y}_T^j \in \mathcal{Y}$, where $\mathcal{Y}$ is the label set, which means that all the domains share the same space of classes. Based on covariate shift and concept drift [21], we aim to learn an

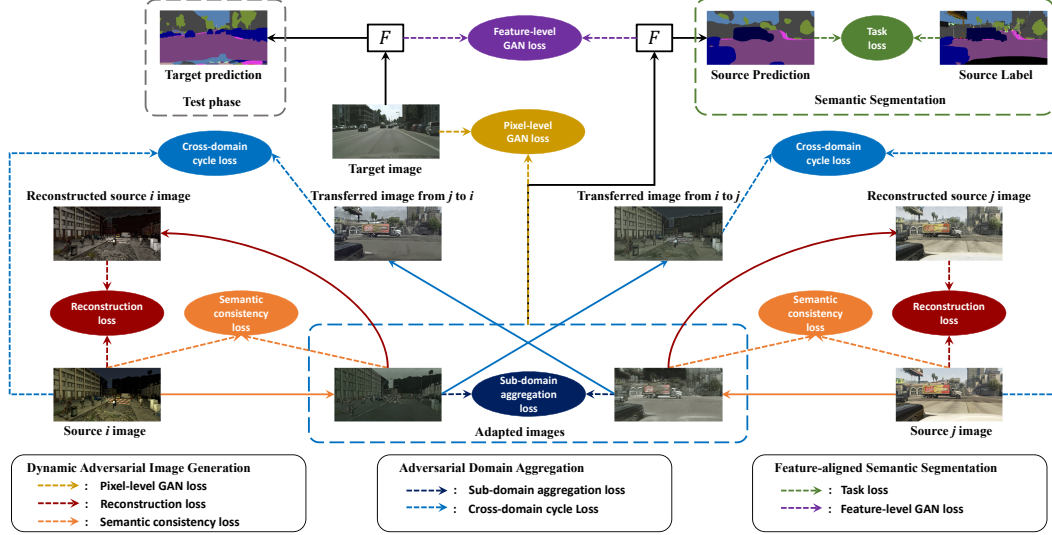

Figure 1: The framework of the proposed Multi-source Adversarial Domain Aggregation Network (MADAN). The colored solid arrows represent generators, while the black solid arrows indicate the segmentation network $F$. The dashed arrows correspond to different losses.

adaptation model that can correctly predict the labels of a sample from the target domain trained on $\{(X_i, Y_i)\}_{i=1}^M$ and $\{X_T\}$.

# 3 Multi-source Adversarial Domain Aggregation Network

In this section, we introduce the proposed Multi-source Adversarial Domain Aggregation Network (MADAN) for semantic segmentation adaptation. The framework is illustrated in Figure 1, which consists of three components: Dynamic Adversarial Image Generation (DAIG), Adversarial Domain Aggregation (ADA), and Feature-aligned Semantic Segmentation (FSS). DAIG aims to generate adapted images from source domains to the target domain from the perspective of visual appearance while preserving the semantic information with a dynamic segmentation model. In order to reduce the distances among the adapted domains and thus generate a more aggregated unified domain, ADA is proposed, including Cross-domain Cycle Discriminator (CCD) and Sub-domain Aggregation Discriminator (SAD). Finally, FSS learns the domain-invariant representations at the feature-level in an adversarial manner. Table 1 compares MADAN with several state-of-the-art DA methods.

## 3.1 Dynamic Adversarial Image Generation

The goal of DAIG is to make images from different source domains visually similar to the target images, as if they are drawn from the same target domain distribution. To this end, for each source domain $S_i$, we introduce a generator $G_{S_i \to T}$ mapping to the target $T$ in order to generate adapted images that fool $D_T$, which is a pixel-level adversarial discriminator. $D_T$ is trained simultaneously with each $G_{S_i \to T}$ to classify real target images $X_T$ from adapted images $G_{S_i \to T}(X_i)$. The corresponding GAN loss function is:

$$\mathcal{L}_{GAN}^{S_i \to T}(G_{S_i \to T}, D_T, X_i, X_T) = \mathbb{E}_{\mathbf{x}_i \sim X_i} \log D_T(G_{S_i \to T}(\mathbf{x}_i)) + \mathbb{E}_{\mathbf{x}_T \sim X_T} \log[1 - D_T(\mathbf{x}_T)]. \quad (1)$$

Since the mapping $G_{S_i \to T}$ is highly under-constrained [36], we employ an inverse mapping $G_{T \to S_i}$ as well as a cycle-consistency loss [39] to enforce $G_{T \to S_i}(G_{S_i \to T}(\mathbf{x}_i)) \approx \mathbf{x}$ and vice versa. Similarly, we introduce $D_i$ to classify $X_i$ from $G_{T \to S_i}(X_T)$, with the following GAN loss:

$$\mathcal{L}_{GAN}^{T \to S_i}(G_{T \to S_i}, D_i, X_T, X_i) = \mathbb{E}_{\mathbf{x}_i \sim X_i} \log[1 - D_i(\mathbf{x}_i)] + \mathbb{E}_{\mathbf{x}_t \sim X_T} \log D_i(G_{T \to S_i}(\mathbf{x}_t)). \quad (2)$$

Table 1: Comparison of the proposed MADAN model with several state-of-the-art domain adaptation methods. The full names of each property from the second to the last columns are pixel-level alignment, feature-level alignment, semantic consistency, cycle consistency, multiple sources, domain aggregation, one task network, and fine-grained prediction, respectively.

| | pixel | feat | sem | cycle | multi | aggr | one | fine |
|---|---|---|---|---|---|---|---|---|
| ADDA [25] | ✗ | ✓ | – | – | ✗ | – | ✓ | ✓ |
| CycleGAN [39] | ✓ | ✗ | ✗ | ✓ | ✗ | – | ✓ | ✗ |
| PiexlDA [37] | ✓ | ✗ | ✗ | ✗ | ✗ | – | ✓ | ✓ |
| SBADA [41] | ✓ | ✗ | ✓ | ✓ | ✗ | – | ✓ | ✗ |
| GTA-GAN [42] | ✓ | ✓ | ✗ | ✗ | ✗ | – | ✓ | ✗ |
| DupGAN [43] | ✓ | ✓ | ✓ | ✗ | ✗ | – | ✓ | ✗ |
| CyCADA [32] | ✓ | ✓ | ✓ | ✓ | ✗ | – | ✓ | ✓ |
| DCTN [68] | ✗ | ✓ | – | – | ✓ | ✗ | ✗ | ✗ |
| MDAN [69] | ✗ | ✓ | – | – | ✓ | ✗ | ✓ | ✗ |
| MMN [70] | ✗ | ✓ | – | – | ✓ | ✗ | ✗ | ✗ |
| MADAN (ours) | ✓ | ✓ | ✓ | ✓ | ✓ | ✓ | ✓ | ✓ |

The cycle-consistency loss [39] ensures that the learned mappings $G_{S_i \to T}$ and $G_{T \to S_i}$ are cycle-consistent, thereby preventing them from contradicting each other, is defined as:

$$\mathcal{L}_{cyc}^{S_i \leftrightarrow T}(G_{S_i \to T}, G_{T \to S_i}, X_i, X_T) = \mathbb{E}_{\mathbf{x}_i \sim X_i} \parallel G_{T \to S_i}(G_{S_i \to T}(\mathbf{x}_i)) - \mathbf{x}_i \parallel_1 +$$
$$\mathbb{E}_{\mathbf{x}_T \sim X_T} \parallel G_{S_i \to T}(G_{T \to S_i}(\mathbf{x}_t)) - \mathbf{x}_t \parallel_1 . \tag{3}$$

The adapted images are expected to contain the same semantic information as original source images, but the semantic consistency is only partially constrained by the cycle consistency loss. The semantic consistency loss in CyCADA [32] was proposed to better preserve semantic information. $\mathbf{x}_i$ and $G_{S_i \to T}(\mathbf{x}_i)$ are both fed into a segmentation model $F_i$ pretrained on $(X_i, Y_i)$. However, since $\mathbf{x}_i$ and $G_{S_i \to T}(\mathbf{x}_i)$ are from different domains, employing the same segmentation model, *i.e.* $F_i$, to obtain the segmentation results and then computing the semantic consistency loss may be detrimental to image generation. Ideally, the adapted images $G_{S_i \to T}(\mathbf{x}_i)$ should be fed into a network $F_T$ trained on the target domain, which is infeasible since target domain labels are not available in UDA. Instead of employing $F_i$ on $G_{S_i \to T}(\mathbf{x}_i)$, we propose to dynamically update the network $F_A$, which takes $G_{S_i \to T}(\mathbf{x}_i)$ as input, so that its optimal input domain (the domain that the network performs best on) gradually changes from that of $F_i$ to $F_T$. We employ the task segmentation model $F$ trained on the adapted domain as $F_A$, *i.e.* $F_A = F$, which has two advantages: (1) $G_{S_i \to T}(\mathbf{x}_i)$ becomes the optimal input domain of $F_A$, and as $F$ is trained to have better performance on the target domain, the semantic loss after $F_A$ would promote $G_{S_i \to T}$ to generate images that are closer to target domain at the pixel-level; (2) since $F_A$ and $F$ can share the parameters, no additional training or memory space is introduced, which is quite efficient. The proposed dynamic semantic consistency (DSC) loss is:

$$\mathcal{L}_{sem}^{S_i}(G_{S_i \to T}, X_i, F_i, F_A) = \mathbb{E}_{\mathbf{x}_i \sim X_i} KL(F_A(G_{S_i \to T}(\mathbf{x}_i)) || F_i(\mathbf{x}_i)), \tag{4}$$

where $KL(\cdot || \cdot)$ is the KL divergence between two distributions.

## 3.2   Adversarial Domain Aggregation

We can train different segmentation models for each adapted domain and combine different predictions with specific weights for target images [68, 70], or we can simply combine all adapted domains together and train one model [69]. In the first strategy, it is challenging to determine how to select the weights for different adapted domains. Moreover, each target image needs to be fed into all segmentation models at reference time, and this is rather inefficient. For the second strategy, since the alignment space is high-dimensional, although the adapted domains are relatively aligned with the target, they may be significantly mis-aligned with each other. In order to mitigate this issue, we propose adversarial domain aggregation to make different adapted domains more closely aggregated with two kinds of discriminators. One is the sub-domain aggregation discriminator (SAD), which is designed to directly make the different adapted domains indistinguishable. For $S_i$, a discriminator $D_A^i$ is introduced with the following loss function:

$$\mathcal{L}_{SAD}^{S_i}(G_{S_1 \to T}, \ldots G_{S_i \to T}, \ldots, G_{S_M \to T}, D_A^i) = \mathbb{E}_{\mathbf{x}_i \sim X_i} \log D_A^i(G_{S_i \to T}(\mathbf{x}_i)) +$$
$$\frac{1}{M-1} \sum_{j \neq i} \mathbb{E}_{\mathbf{x}_j \sim X_j} \log[1 - D_A^i(G_{S_j \to T}(\mathbf{x}_j))]. \tag{5}$$

The other is the cross-domain cycle discriminator (CCD). For each source domain $S_i$, we transfer the images from the adapted domains $G_{S_j \to T}(X_j)$, $j = 1, \cdots, M, j \neq i$ back to $S_i$ using $G_{T \to S_i}$ and employ the discriminator $D_i$ to classify $X_i$ from $G_{T \to S_i}(G_{S_j \to T}(X_j))$, which corresponds to the following loss function:

$$\mathcal{L}_{CCD}^{S_i}(G_{T \to S_1}, \dots G_{T \to S_{i-1}}, G_{T \to S_{i+1}}, \dots, G_{T \to S_M}, G_{S_i \to T}, D_i) = \mathbb{E}_{\mathbf{x}_i \sim X_i} \log D_i(\mathbf{x}_i) +$$
$$\frac{1}{M-1} \sum\nolimits_{j \neq i} \mathbb{E}_{\mathbf{x}_j \sim X_j} \log[1 - D_i(G_{T \to S_i}((G_{S_j \to T}(\mathbf{x}_j))))]. \tag{6}$$

Please note that using a more sophisticated combination of different discriminators' losses to better aggregate the domains with larger distances might improve the performance. We leave this as future work and would explore this direction by dynamic weighting of the loss terms and incorporating some prior domain knowledge of the sources.

### 3.3 Feature-aligned Semantic Segmentation

After adversarial domain aggregation, the adapted images of different domains $X_i'(i = 1, \cdots, M)$ are more closely aggregated and aligned. Meanwhile, the semantic consistency loss in dynamic adversarial image generation ensures that the semantic information, *i.e.* the segmentation labels, is preserved before and after image translation. Suppose the images of the unified aggregated domain are $X' = \bigcup_{i=1}^{M} X_i'$ and corresponding labels are $Y = \bigcup_{i=1}^{M} Y_i$. We can then train a task segmentation model $F$ based on $X'$ and $Y$ with the following cross-entropy loss:

$$\mathcal{L}_{task}(F, X', Y) = -\mathbb{E}_{(\mathbf{x}', \mathbf{y}) \sim (X', Y)} \sum\nolimits_{l=1}^{L} \sum\nolimits_{h=1}^{H} \sum\nolimits_{w=1}^{W} \mathbb{1}_{[l = \mathbf{y}_{h,w}]} \log(\sigma(F_{l,h,w}(\mathbf{x}'))), \tag{7}$$

where $L$ is the number of classes, $H, W$ are the height and width of the adapted images, $\sigma$ is the softmax function, $\mathbb{1}$ is an indicator function, and $F_{l,h,w}(\mathbf{x}')$ is the value of $F(\mathbf{x}')$ at index $(l, h, w)$.

Further, we impose a feature-level alignment between $X'$ and $X_T$, which can improve the segmentation performance during inference of $X_T$ on the segmentation model $F$. We introduce a discriminator $D_F$ to achieve this goal. The feature-level GAN loss is defined as:

$$\mathcal{L}_{feat}(F_f, D_{F_f}, X', X_T) = \mathbb{E}_{\mathbf{x}' \sim X'} \log D_{F_f}(F_f(\mathbf{x}')) + \mathbb{E}_{\mathbf{x}_T \sim X_T} \log[1 - D_{F_f}(F_f(\mathbf{x}_T))], \tag{8}$$

where $F_f(\cdot)$ is the output of the last convolution layer (*i.e.* a feature map) of the encoder in $F$.

### 3.4 MADAN Learning

The proposed MADAN learning framework utilizes adaptation techniques including pixel-level alignment, cycle-consistency, semantic consistency, domain aggregation, and feature-level alignment. Combining all these components, the overall objective loss function of MADAN is:

$$\mathcal{L}_{MADAN}(G_{S_1 \to T} \cdots G_{S_M \to T}, G_{T \to S_1} \cdots G_{T \to S_M}, D_1 \cdots D_M, D_A^1 \cdots D_A^M, D_{F_f}, F)$$
$$= \sum\nolimits_i \Big[ \mathcal{L}_{GAN}^{S_i \to T}(G_{S_i \to T}, D_T, X_i, X_T) + \mathcal{L}_{GAN}^{T \to S_i}(G_{T \to S_i}, D_i, X_T, X_i)$$
$$+ \mathcal{L}_{cyc}^{S_i \leftrightarrow T}(G_{S_i \to T}, G_{T \to S_i}, X_i, X_T) + \mathcal{L}_{sem}^{S_i}(G_{S_i \to T}, X_i, F_i, F)$$
$$+ \mathcal{L}_{SAD}^{S_i}(G_{S_1 \to T}, \dots G_{S_i \to T}, \dots, G_{S_M \to T}, D_A^i)$$
$$+ \mathcal{L}_{CCD}^{S_i}(G_{T \to S_1}, \dots G_{T \to S_{i-1}}, G_{T \to S_{i+1}}, \dots, G_{T \to S_M}, G_{S_i \to T}, D_i) \Big]$$
$$+ \mathcal{L}_{task}(F, X', Y) + \mathcal{L}_{feat}(F_f, D_{F_f}, X', X_T). \tag{9}$$

The training process corresponds to solving for a target model $F$ according to the optimization:

$$F^* = \arg \min_F \min_D \max_G \mathcal{L}_{MADAN}(G, D, F), \tag{10}$$

where $G$ and $D$ represent all the generators and discriminators in Eq. (9), respectively.

## 4 Experiments

In this section, we first introduce the experimental settings and then compare the segmentation results of the proposed MADAN and several state-of-the-art approaches both quantitatively and qualitatively, followed by some ablation studies.

Table 2: Comparison with the state-of-the-art DA methods for semantic segmentation from GTA and SYNTHIA to Cityscapes. The best class-wise IoU and mIoU trained on the source domains are emphasized in bold (similar below).

| Standards | Method | road | sidewalk | building | wall | fence | pole | t-light | t-sign | vegetation | sky | person | rider | car | bus | m-bike | bicycle | mIoU |
|---|---|---|---|---|---|---|---|---|---|---|---|---|---|---|---|---|---|---|
| Source-only | GTA | 54.1 | 19.6 | 47.4 | 3.3 | 5.2 | 3.3 | 0.5 | 3.0 | 69.2 | 43.0 | 31.3 | 0.1 | 59.3 | 8.3 | 0.2 | 0.0 | 21.7 |
| | SYNTHIA | 3.9 | 14.5 | 45.0 | 0.7 | 0.0 | 14.6 | 0.7 | 2.6 | 68.2 | 68.4 | 31.5 | 4.6 | 31.5 | 7.4 | 0.3 | 1.4 | 18.5 |
| | GTA+SYNTHIA | 44.0 | 19.0 | 60.1 | 11.1 | 13.7 | 10.1 | 5.0 | 4.7 | 74.7 | 65.3 | 40.8 | 2.3 | 43.0 | 15.9 | 1.3 | 1.4 | 25.8 |
| GTA-only DA | FCN Wld [47] | 70.4 | 32.4 | 62.1 | 14.9 | 5.4 | 10.9 | 14.2 | 2.7 | 79.2 | 64.6 | 44.1 | 4.2 | 70.4 | 7.3 | 3.5 | 0.0 | 27.1 |
| | CDA [48] | 74.8 | 22.0 | 71.7 | 6.0 | 11.9 | 8.4 | 16.3 | 11.1 | 75.7 | 66.5 | 38.0 | 9.3 | 55.2 | 18.9 | 16.8 | 14.6 | 28.9 |
| | ROAD [50] | 85.4 | 31.2 | 78.6 | **27.9** | **22.2** | 21.9 | 23.7 | 11.4 | 80.7 | 68.9 | 48.5 | 14.1 | 78.0 | 23.8 | 8.3 | 0.0 | 39.0 |
| | AdaptSeg [71] | **87.3** | 29.8 | 78.6 | 21.1 | 18.2 | 22.5 | 21.5 | 11.0 | 79.7 | 71.3 | 46.8 | 6.5 | **80.1** | 26.9 | 10.6 | 0.3 | 38.3 |
| | CyCADA [32] | 85.2 | 37.2 | 76.5 | 21.8 | 15.0 | 23.8 | 22.9 | 21.5 | 80.5 | 60.7 | 50.5 | 9.0 | 76.9 | 28.2 | 4.5 | 0.0 | 38.7 |
| | DCAN [55] | 82.3 | 26.7 | 77.4 | 23.7 | 20.5 | 20.4 | **30.3** | 15.9 | 80.9 | 69.5 | **52.6** | 11.1 | 79.6 | 21.2 | 17.0 | 6.7 | 39.8 |
| SYNTHIA-only DA | FCN Wld [47] | 11.5 | 19.6 | 30.8 | 4.4 | 0.0 | 20.3 | 0.1 | 11.7 | 42.3 | 68.7 | 51.2 | 3.8 | 54.0 | 3.2 | 0.2 | 0.6 | 20.2 |
| | CDA [48] | 65.2 | 26.1 | 74.9 | 0.1 | 0.5 | 10.7 | 3.7 | 3.0 | 76.1 | 70.6 | 47.1 | 8.2 | 43.2 | 20.7 | 0.7 | 13.1 | 29.0 |
| | ROAD [50] | 77.7 | 30.0 | 77.5 | 9.6 | 0.3 | **25.8** | 10.3 | 15.6 | 77.6 | **79.8** | 44.5 | 16.6 | 67.8 | 14.5 | 7.0 | 23.8 | 36.2 |
| | CyCADA [32] | 66.2 | 29.6 | 65.3 | 0.5 | 0.2 | 15.1 | 4.5 | 6.9 | 67.1 | 68.2 | 42.8 | 14.1 | 51.2 | 12.6 | 2.4 | 20.7 | 29.2 |
| | DCAN [55] | 79.9 | 30.4 | 70.8 | 1.6 | 0.6 | 22.3 | 6.7 | **23.0** | 76.9 | 73.9 | 41.9 | 16.7 | 61.7 | 11.5 | 10.3 | **38.6** | 35.4 |
| Source-combined DA | CyCADA [32] | 82.8 | 35.8 | 78.2 | 17.5 | 15.1 | 10.8 | 6.1 | 19.4 | 78.6 | 77.2 | 44.5 | 15.3 | 74.9 | 17.0 | 10.3 | 12.9 | 37.3 |
| Multi-source DA | MDAN [69] | 64.2 | 19.7 | 63.8 | 13.1 | 19.4 | 5.5 | 5.2 | 6.8 | 71.6 | 61.1 | 42.0 | 12.0 | 62.7 | 2.9 | 12.3 | 8.1 | 29.4 |
| | **MADAN (Ours)** | 86.2 | **37.7** | **79.1** | 20.1 | 17.8 | 15.5 | 14.5 | 21.4 | 78.5 | 73.4 | 49.7 | **16.8** | 77.8 | **28.3** | **17.7** | 27.5 | **41.4** |
| Oracle-Train on Tgt | FCN [5] | 96.4 | 74.5 | 87.1 | 35.3 | 37.8 | 36.4 | 46.9 | 60.1 | 89.0 | 89.8 | 65.6 | 35.9 | 76.9 | 64.1 | 40.5 | 65.1 | 62.6 |

Table 3: Comparison with the state-of-the-art DA methods for semantic segmentation from GTA and SYNTHIA to BDDS. The best class-wise IoU and mIoU are emphasized in bold.

| Standards | Method | road | sidewalk | building | wall | fence | pole | t-light | t-sign | vegetation | sky | person | rider | car | bus | m-bike | bicycle | mIoU |
|---|---|---|---|---|---|---|---|---|---|---|---|---|---|---|---|---|---|---|
| Source-only | GTA | 50.2 | 18.0 | 55.1 | 3.1 | 7.8 | 7.0 | 0.0 | 3.5 | 61.0 | 50.4 | 19.2 | 0.0 | 58.1 | 3.2 | **19.8** | 0.0 | 22.3 |
| | SYNTHIA | 7.0 | 6.0 | 50.5 | 0.0 | 0.0 | 15.1 | 0.2 | 2.4 | 60.3 | **85.6** | 16.5 | 0.5 | 36.7 | 3.3 | 0.0 | 3.5 | 17.1 |
| | GTA+SYNTHIA | 54.5 | 19.6 | 64.0 | 3.2 | 3.6 | 5.2 | 0.0 | 0.0 | 61.3 | 82.2 | 13.9 | 0.0 | 55.5 | 16.7 | 13.4 | 0.0 | 24.6 |
| GTA-only DA | CyCADA [32] | **77.9** | 26.8 | 68.8 | 13.0 | **19.7** | 13.5 | **18.2** | **22.3** | 64.2 | 84.2 | 39.0 | **22.6** | **72.0** | 11.5 | 15.9 | 2.0 | 35.7 |
| SYNTHIA-only DA | CyCADA [32] | 55 | 13.8 | 45.2 | 0.1 | 0.0 | 13.2 | 0.5 | 10.6 | 63.3 | 67.4 | 22.0 | 6.9 | 52.5 | 10.5 | 10.4 | 13.3 | 24.0 |
| Source-combined DA | CyCADA [32] | 61.5 | 27.6 | **72.1** | 6.5 | 2.8 | 15.7 | 10.8 | 18.1 | 78.3 | 73.8 | 44.9 | 16.3 | 41.5 | 21.1 | 21.8 | **25.9** | 33.7 |
| Multi-source DA | MDAN [69] | 35.9 | 15.8 | 56.9 | 5.8 | 16.3 | 9.5 | 8.6 | 6.2 | 59.1 | 80.1 | 24.5 | 9.9 | 53.8 | 11.8 | 2.9 | 1.6 | 25.0 |
| | **MADAN (Ours)** | 60.2 | **29.5** | 66.6 | **16.9** | 10.0 | **16.6** | 10.9 | 16.4 | **78.8** | 75.1 | **47.5** | 17.3 | 48.0 | **24.0** | 13.2 | 17.3 | **36.3** |
| Oracle-Train on Tgt | FCN [5] | 91.7 | 54.7 | 79.5 | 25.9 | 42.0 | 23.6 | 30.9 | 34.6 | 81.2 | 91.6 | 49.6 | 23.5 | 85.4 | 64.2 | 28.4 | 41.1 | 53.0 |

## 4.1 Experimental Settings

**Datasets**. In our adaptation experiments, we use synthetic GTA [18] and SYNTHIA [19] datasets as the source domains and real Cityscapes [15] and BDDS [72] datasets as the target domains.

**Baselines**. We compare MADAN with the following methods. **(1) Source-only**, *i.e.* train on the source domains and test on the target domain directly. We can view this as a lower bound of DA. **(2) Single-source DA**, perform multi-source DA via single-source DA, including FCNs Wld [47], CDA [48], ROAD [50], AdaptSeg [71], CyCADA [32], and DCAN [55]. **(3) Multi-source DA**, extend some single-source DA method to multi-source settings, including MDAN [69]. For comparison, we also report the results of an oracle setting, where the segmentation model is both trained and tested on the target domain. For the source-only and single-source DA standards, we employ two strategies: (1) single-source, *i.e.* performing adaptation on each single source; (2) source-combined, *i.e.* all source domains are combined into a traditional single source. For MDAN, we extend the original classification network for our segmentation task.

**Evaluation Metric**. Following [47, 48, 32, 56], we employ mean intersection-over-union (mIoU) to evaluate the segmentation results. In the experiments, we take the 16 intersection classes of GTA and SYNTHIA, compatible with Cityscapes and BDDS, for all mIoU evaluations.

**Implementation Details**. Although MADAN could be trained in an end-to-end manner, due to constrained hardware resources, we train it in three stages. First, we train two CycleGANs (9 residual blocks for generator and 4 convolution layers for discriminator) [39] without semantic consistency loss, and then train an FCN $F$ on the adapted images with corresponding labels from the source domains. Second, after updating $F_A$ with $F$ trained above, we generate adapted images using CycleGAN with the proposed DSC loss in Eq. (4) and aggregate different adapted domains using SAD and CCD. Finally, we train an FCN on the newly adapted images in the aggregated domain with feature-level alignment. The above stages are trained iteratively.

We choose to use FCN [5] as our semantic segmentation network, and, as the VGG family of networks is commonly used in reporting DA results, we use VGG-16 [73] as the FCN backbone. The weights

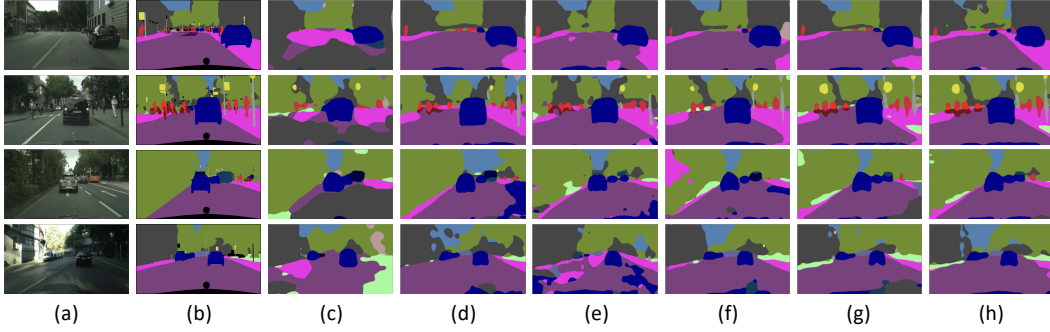

Figure 2: Qualitative semantic segmentation result from GTA and SYNTHIA to Cityscapes. From left to right are: (a) original image, (b) ground truth annotation, (c) source only from GTA, (d) CycleGANs on GTA and SYNTHIA, (e) +CCD+DSC, (f) +SAD+DSC, (g) +CCD+SAD+DSC, and (h) +CCD+SAD+DSC+Feat (MADAN).

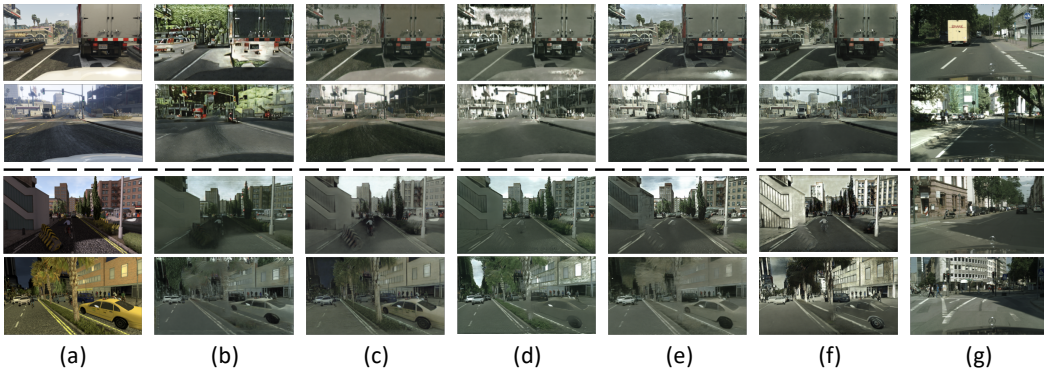

Figure 3: Visualization of image translation. From left to right are: (a) original source image, (b) CycleGAN, (c) CycleGAN+DSC, (d) CycleGAN+CCD+DSC, (e) CycleGAN+SAD+DSC, (f) CycleGAN+CCD+SAD+DSC, and (g) target Cityscapes image. The top two rows and bottom rows are GTA → Cityscapes and SYNTHIA → Cityscapes, respectively.

of the feature extraction layers in the networks are initialized from models trained on ImageNet [74]. The network is implemented in PyTorch and trained with Adam optimizer [75] using a batch size of 8 with initial learning rate 1e-4. All the images are resized to $600 \times 1080$, and are then cropped to $400 \times 400$ during the training of the pixel-level adaptation for 20 epochs. SAD and CCD are frozen in the first 5 and 10 epochs, respectively.

## 4.2 Comparison with State-of-the-art

The performance comparisons between the proposed MADAN model and the other baselines, including source-only, single-source DA, and multi-source DA, as measured by class-wise IoU and mIoU are shown in Table 2 and Table 3. From the results, we have the following observations:

**(1)** The source-only method that directly transfers the segmentation models trained on the source domains to the target domain obtains the worst performance in most adaptation settings. This is obvious, because the joint probability distributions of observed images and labels are significantly different among the sources and the target, due to the presence of domain shift. Without domain adaptation, the direct transfer cannot well handle this domain gap. Simply combining different source domains performs better than each single source, which indicates the superiority of multiple sources over single source despite the domain shift among different sources.

**(2)** Comparing source-only with single-source DA respectively on GTA and SYNTHIA, it is clear that all adaptation methods perform better, which demonstrates the effectiveness of domain adaptation in semantic segmentation. Comparing the results of CyCADA in single-source and source-combined

Table 4: Comparison between the proposed dynamic semantic consistency (DSC) loss in MADAN and the original SC loss in [32] on Cityscapes. The better mIoU for each pair is emphasized in bold.

| Source | Method | road | sidewalk | building | wall | fence | pole | t-light | t-sign | vegetation | sky | person | rider | car | bus | m-bike | bicycle | mIoU |
|---|---|---|---|---|---|---|---|---|---|---|---|---|---|---|---|---|---|---|
| GTA | CycleGAN+SC | 85.6 | 30.7 | 74.7 | 14.4 | 13.0 | 17.6 | 13.7 | 5.8 | 74.6 | 69.9 | 38.2 | 3.5 | 72.3 | 5.0 | 3.6 | 0.0 | 32.7 |
| | CycleGAN+DSC | 76.6 | 26.0 | 76.3 | 17.3 | 18.8 | 13.6 | 13.2 | 17.9 | 78.8 | 63.9 | 47.4 | 14.8 | 72.2 | 24.1 | 19.8 | 10.8 | **38.1** |
| | CyCADA w/ SC | 85.2 | 37.2 | 76.5 | 21.8 | 15.0 | 23.8 | 21.5 | 22.9 | 80.5 | 60.7 | 50.5 | 9.0 | 76.9 | 28.2 | 9.8 | 0.0 | 38.7 |
| | CyCADA w/ DSC | 84.1 | 27.3 | 78.3 | 21.6 | 18.0 | 13.8 | 14.1 | 16.7 | 78.1 | 66.9 | 47.8 | 15.4 | 78.7 | 23.4 | 22.3 | 14.4 | **40.0** |
| SYNTHIA | CycleGAN+SC | 64.0 | 29.4 | 61.7 | 0.3 | 0.1 | 15.3 | 3.4 | 5.0 | 63.4 | 68.4 | 39.4 | 11.5 | 46.6 | 10.4 | 2.0 | 16.4 | 27.3 |
| | CycleGAN + DSC | 68.4 | 29.0 | 65.2 | 0.6 | 0.0 | 15.0 | 0.1 | 4.0 | 75.1 | 70.6 | 45.0 | 11.0 | 54.9 | 18.2 | 3.9 | 26.7 | **30.5** |
| | CyCADA w/ SC | 66.2 | 29.6 | 65.3 | 0.5 | 0.2 | 15.1 | 4.5 | 6.9 | 67.1 | 68.2 | 42.8 | 14.1 | 51.2 | 12.6 | 2.4 | 20.7 | 29.2 |
| | CyCADA w/ DSC | 69.8 | 27.2 | 68.5 | 5.8 | 0.0 | 11.6 | 0.0 | 2.8 | 75.7 | 58.3 | 44.3 | 10.5 | 68.1 | 22.1 | 11.8 | 32.7 | **31.8** |

Table 5: Comparison between the proposed dynamic semantic consistency (DSC) loss in MADAN and the original SC loss in [32] on BDDS. The better mIoU for each pair is emphasized in bold.

| Source | Method | road | sidewalk | building | wall | fence | pole | t-light | t-sign | vegetation | sky | person | rider | car | bus | m-bike | bicycle | mIoU |
|---|---|---|---|---|---|---|---|---|---|---|---|---|---|---|---|---|---|---|
| GTA | CycleGAN+SC | 62.1 | 20.9 | 59.2 | 6.0 | 23.5 | 12.8 | 9.2 | 22.4 | 65.9 | 78.4 | 34.7 | 11.4 | 64.4 | 14.2 | 10.9 | 1.9 | 31.1 |
| | CycleGAN+DSC | 74.4 | 23.7 | 65.0 | 8.6 | 17.2 | 10.7 | 14.2 | 19.7 | 59.0 | 82.8 | 36.3 | 19.6 | 69.7 | 4.3 | 17.6 | 4.2 | **32.9** |
| | CyCADA w/ SC | 68.8 | 23.7 | 67.0 | 7.5 | 16.2 | 9.4 | 11.3 | 22.2 | 60.5 | 82.1 | 36.1 | 20.6 | 63.2 | 15.2 | 16.6 | 3.4 | 32.0 |
| | CyCADA w/ DSC | 70.5 | 32.4 | 68.2 | 10.5 | 17.3 | 18.4 | 16.6 | 21.8 | 65.6 | 82.2 | 38.1 | 16.1 | 73.3 | 20.8 | 12.6 | 3.7 | **35.5** |
| SYNTHIA | CycleGAN+SC | 50.6 | 13.6 | 50.5 | 0.2 | 0.0 | 7.9 | 0.0 | 0.0 | 63.8 | 58.3 | 21.6 | 7.8 | 50.2 | 1.8 | 2.2 | 19.9 | 21.8 |
| | CycleGAN + DSC | 57.3 | 13.4 | 56.1 | 2.7 | 14.1 | 9.8 | 7.7 | 17.1 | 65.5 | 53.1 | 11.4 | 1.4 | 51.4 | 13.9 | 3.9 | 8.7 | **22.5** |
| | CyCADA w/ SC | 49.5 | 11.1 | 46.6 | 0.7 | 0.0 | 10.0 | 0.4 | 7.0 | 61.0 | 74.6 | 17.5 | 7.2 | 50.9 | 5.8 | 13.1 | 4.3 | 23.4 |
| | CyCADA w/ DSC | 55 | 13.8 | 45.2 | 0.1 | 0.0 | 13.2 | 0.5 | 10.6 | 63.3 | 67.4 | 22.0 | 6.9 | 52.5 | 10.5 | 10.4 | 13.3 | **24.0** |

settings, we can conclude that simply combining different source domains and performing single-source DA may result in performance degradation.

**(3)** MADAN achieves the highest mIoU score among all adaptation methods, and benefits from the joint consideration of pixel-level and feature-level alignments, cycle-consistency, dynamic semantic-consistency, domain aggregation, and multiple sources. MADAN also significantly outperforms source-combined DA, in which domain shift also exists among different sources. By bridging this gap, multi-source DA can boost the adaptation performance. On the one hand, compared to single-source DA [47, 48, 50, 71, 32, 55], MADAN utilizes more useful information from multiple sources. On the other hand, other multi-source DA methods [68, 69, 70] only consider feature-level alignment, which may be enough for course-grained tasks, *e.g.* image classification, but is obviously insufficient for fine-grained tasks, *e.g.* semantic segmentation, a pixel-wise prediction task. In addition, we consider pixel-level alignment with a dynamic semantic consistency loss and further aggregate different adapted domains.

**(4)** The oracle method that is trained on the target domain performs significantly better than the others. However, to train this model, the ground truth segmentation labels from the target domain are required, which are actually unavailable in UDA settings. We can deem this performance as a upper bound of UDA. Obviously, a large performance gap still exists between all adaptation algorithms and the oracle method, requiring further efforts on DA.

**Visualization.** The qualitative semantic segmentation results are shown in Figure 2. We can clearly see that after adaptation by the proposed method, the visual segmentation results are improved notably. We also visualize the results of pixel-level alignment from GTA and SYNTHIA to Cityscapes in Figure 3. We can see that with our final proposed pixel-level alignment method (f), the styles of the images are close to Cityscapes while the semantic information is well preserved.

### 4.3 Ablation Study

First, we compare the proposed dynamic semantic consistency (DSC) loss in MADAN with the original semantic consistency (SC) loss in CyCADA [32]. As shown in Table 4 and Table 5, we can see that for all simulation to real adaptations, DSC achieves better results. After demonstrating its value, we employ the DSC loss in subsequent experiments.

Second, we incrementally investigate the effectiveness of different components in MADAN on both Cityscapes and BDDS. The results are shown in Table 6 and Table 7. We can observe that: (1) both domain aggregation methods, *i.e.* SAD and CCD, can obtain better performance by making different adapted domains more closely aggregated, while SAD outperforms CCD; (2) adding the

Table 6: Ablation study on different components in MADAN on Cityscapes. Baseline denotes using piexl-level alignment with cycle-consistency, +SAD denotes using the sub-domain aggregation discriminator, +CCD denotes using the cross-domain cycle discriminator, +DSC denotes using the dynamic semantic consistency loss, and +Feat denotes using feature-level alignment.

| Method | road | sidewalk | building | wall | fence | pole | t-light | t-sign | vegetation | sky | person | rider | car | bus | m-bike | bicycle | mIoU |
|---|---|---|---|---|---|---|---|---|---|---|---|---|---|---|---|---|---|
| Baseline | 74.9 | 27.6 | 67.5 | 9.1 | 10.0 | 12.8 | 1.4 | 13.6 | 63.0 | 47.1 | 41.7 | 13.5 | 60.8 | 22.4 | 6.0 | 8.1 | 30.0 |
| +SAD | 79.7 | 33.2 | 75.9 | 11.8 | 3.6 | 15.9 | 8.6 | 15.0 | 74.7 | **78.9** | 44.2 | 17.1 | 68.2 | 24.9 | 16.7 | 14.0 | 36.4 |
| +CCD | 82.1 | 36.3 | 69.8 | 9.5 | 4.9 | 11.8 | 12.5 | 15.3 | 61.3 | 54.1 | 49.7 | 10.0 | 70.7 | 9.7 | 19.7 | 12.4 | 33.1 |
| +SAD+CCD | 82.7 | 35.3 | 76.5 | 15.4 | **19.4** | 14.1 | 7.2 | 13.9 | 75.3 | 74.2 | **50.9** | **19.0** | 66.5 | 26.6 | 16.3 | 6.7 | 37.5 |
| +SAD+DSC | 83.1 | 36.6 | 78.0 | **23.3** | 12.6 | 11.8 | 3.5 | 11.3 | 75.5 | 74.8 | 42.2 | 17.9 | 72.2 | 27.2 | 13.8 | 10.0 | 37.1 |
| +CCD+DSC | **86.8** | 36.9 | 78.6 | 16.2 | 8.1 | **17.7** | 8.9 | 13.7 | 75.0 | 74.8 | 42.2 | 18.2 | 74.6 | 22.5 | **22.9** | 12.7 | 38.1 |
| +SAD+CCD+DSC | 84.2 | 35.1 | 78.7 | 17.1 | 18.7 | 15.4 | **15.7** | **24.1** | 77.9 | 72.0 | 49.2 | 17.1 | 75.2 | 24.1 | 18.9 | 19.2 | 40.2 |
| +SAD+CCD+DSC+Feat | 86.2 | **37.7** | **79.1** | 20.1 | 17.8 | 15.5 | 14.5 | 21.4 | **78.5** | 73.4 | 49.7 | 16.8 | **77.8** | **28.3** | 17.7 | **27.5** | **41.4** |

Table 7: Ablation study on different components in MADAN on BDDS.

| Method | road | sidewalk | building | wall | fence | pole | t-light | t-sign | vegetation | sky | person | rider | car | bus | m-bike | bicycle | mIoU |
|---|---|---|---|---|---|---|---|---|---|---|---|---|---|---|---|---|---|
| Baseline | 31.3 | 17.4 | 55.4 | 2.6 | 12.9 | 12.4 | 6.5 | 18.0 | 63.2 | 79.9 | 21.2 | 5.6 | 44.1 | 14.2 | 6.1 | 11.7 | 24.6 |
| +SAD | 58.9 | 18.7 | 61.8 | 6.4 | 10.7 | 17.1 | 20.3 | 17.0 | 67.3 | 83.7 | 21.1 | 6.7 | 66.6 | 22.7 | 4.5 | 14.9 | 31.2 |
| +CCD | 52.7 | 13.6 | 63.0 | 6.6 | 11.2 | 17.8 | **21.5** | **18.9** | 67.4 | **84.0** | 9.2 | 2.2 | 63.0 | 21.6 | 2.0 | 14.0 | 29.3 |
| +SAD+CCD | 61.6 | 20.2 | 61.7 | 7.2 | 12.1 | **18.5** | 19.8 | 16.7 | 64.2 | 83.2 | 25.9 | 7.3 | 66.8 | 22.2 | 5.3 | 14.9 | 31.8 |
| +SAD+DSC | 60.2 | 29.5 | 66.6 | 16.9 | 10.0 | 16.6 | 10.9 | 16.4 | **78.8** | 75.1 | **47.5** | 17.3 | 48.0 | **24.0** | 13.2 | **17.3** | 34.3 |
| +CCD+DSC | 61.5 | 27.6 | 72.1 | 6.5 | 12.8 | 15.7 | 10.8 | 18.1 | 78.3 | 73.8 | 44.9 | 16.3 | 41.5 | 21.1 | 21.8 | 15.9 | 33.7 |
| +SAD+CCD+DSC | 64.6 | **38.0** | 75.8 | 17.8 | 13.0 | 9.8 | 5.9 | 4.6 | 74.8 | 76.9 | 41.8 | **24.0** | **69.0** | 20.4 | 23.7 | 11.3 | 35.3 |
| +SAD+CCD+DSC+Feat | **69.1** | 36.3 | **77.9** | **21.5** | **17.4** | 13.8 | 4.1 | 16.2 | 76.5 | 76.2 | 42.2 | 16.4 | 56.3 | 22.4 | **24.5** | 13.5 | **36.3** |

DSC loss could further improve the mIoU score, again demonstrating the effectiveness of DSC; (3) feature-level alignments also contribute to the adaptation task; (4) the modules are orthogonal to each other to some extent, since adding each one of them does not introduce performance degradation.

## 4.4 Discussions

**Computation cost.** Since the proposed framework deals with a harder problem, *i.e.* multi-source domain adaptation, more modules are used to align different sources, which results in a larger model. In our experiments, MADAN is trained on 4 NVIDIA Tesla P40 GPUs for 40 hours using two source domains which is about twice the training time as on a single source. However, MADAN does not introduce any additional computation during inference, which is the biggest concern in real industrial applications, *e.g.* autonomous driving.

**On the poorly performing classes.** There are two main reasons for the poor performance on certain classes (*e.g.* fence and pole): 1) lack of images containing these classes and 2) structural differences of objects between simulation images and real images (*e.g.* the trees in simulation images are much taller than those in real images). Generating more images for different classes and improving the diversity of objects in the simulation environment are two promising directions for us to explore in future work that may help with these problems.

## 5 Conclusion

In this paper, we studied multi-source domain adaptation for semantic segmentation from synthetic data to real data. A novel framework, termed Multi-source Adversarial Domain Aggregation Network (MADAN), is designed with three components. For each source domain, we generated adapted images with a novel dynamic semantic consistency loss. Further we proposed a sub-domain aggregation discriminator and cross-domain cycle discriminator to better aggregate different adapted domains. Together with other techniques such as pixel- and feature-level alignments as well as cycle-consistency, MADAN achieves 15.6%, 1.6%, 4.1%, and 12.0% mIoU improvements compared with best source-only, best single-source DA, source-combined DA, and other multi-source DA, respectively on Cityscapes from GTA and SYNTHIA, and 11.7%, 0.6%, 2.6%, 11.3% on BDDS. For further studies, we plan to investigate multi-modal DA, such as using both image and LiDAR data, to better boost the adaptation performance. Improving the computational efficiency of MADAN, with techniques such as neural architecture search, is another direction worth investigating.

**Acknowledgments**

This work is supported by Berkeley DeepDrive and the National Natural Science Foundation of China (No. 61701273).

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
