[Reviews · NeurIPS 2019]

Reviewer 1



- The paper extends the unsupervised domain adaptation task for semantic segmentation to multiple sources. This seems to be the first paper that does that. - The paper is well written and sufficiently clear. - The results are sufficient and convincing. - The approach can be considered as reasonable extension of CyCADA to leverage multiple sources. There is also an improvement of CyCADA given by a new dynamic semantic segmentation loss. Besides that, the other major novelty is the introduction of losses to produce an adversarial domain aggregation. - The most important aspect of the paper is that it builds on solid previous work and goes into a new direction, which is to use multiple sources for adaptation of segmentation networks. - No major concerns, just that the Authors have not done much to describe the details of the architecture used. How are D_T and D_i defined? What network structures have you used for them and also for the generator networks? - In Eq(9) you may want to replace F_A with F.

Reviewer 2



This paper proposes to leverage training data from multiple source domains to strengthen the ability of segmentation models on processing domain shift and dataset bias. It well reviews the literature of related work and clearly points out the differences between this paper and the prior ones. The presentation is good and the ideas are also interesting and original. Nevertheless, I still think this paper has the following issues that need to be settled. 1) The paper aims to handle the domain shift problem by leveraging multiple source domain datasets. Intuitively, this would definitely strengthen the segmentation ability of the resulting model compared to those trained on a single source dataset. Despite so, I think it is still necessary to experimentally demonstrate that multiple sources indeed assist compared to the case where only one source dataset is provided under the setting that all other configurations are kept unchanged. However, I do not see any results in this paper about this. In Table 2, the authors do list results based on single source domain but the experiment environment is different. I think this experiment would more convincingly explain that multiple sources help. 2) I suggest that the authors should re-organize the layout of Figure 1. It really confuses me a lot and I think this will also happen to other readers. It really takes me some time to figure out what all these arrow (in various colors and with different styles) are used for. 3) The adversarial domain aggregation part seems naive a bit. Loss functions for different discriminators are just simply linearly combined tegother. This is too straightforward in my view. I originally thought there would be some discussion on how to design this part and why doing so but no such content is found.

Reviewer 3



I am not entirely convinced about the validity of incorporating semantic alignment, based on the assumption that the aggregated domain image and the target image should have the same segmentation labels, which seems rather strong over a diverse source-target pair. It is unclear what exactly make a "feature" - this needs to be clarified so as to justify the proposed feature alignment. Overall, the framework reflected by Eq. 9 is rather large, and it is unclear how costly it is computationally. As for the experiment results, two questions can be asked: - Compared with the source-combine DA, the proposed MADAN framework gives only marginal improvements on most of the classes. Interestingly, for "Sky", source-combine DA performs much better. Is there any explanation? On the other hand, poorly performing classes remain so with MADAN - is it really worthwhile to use such a complicated framework? - The ablation study results are not very convincing. Using SAD alone achieves the best performance on "sky"; further progressive additions receive poorer results. In fact, contrary to the claims made by the author, progressive additions actually degrade the performance, e.g. for 5 classes, SAD+CCD+DSC+Feat gives worse results compared with SAD+CCD+DSC. SAD+CCD gives the best result on "person", better than SAD+CCD+DSC and SAD+CCD+DSC+Feat. Although the addition seems to give growing mIoU indices, to me the often marginal improvements obtained by these algorithmic components may not be counted any significant contribution. Section 4.4 gives some generalisation results. It could be strengthened by adding a progressive multi-source DA process, from GTA/Synthia->BSDS, to GTA/Synthia/Cityscapes->BSDS, so as to demonstrate a better case for "multi-source". Some minor corrections: - Eq.3, missing a half parenthesis after G_S_i->T(x_i)-x_i: should be G_S_i(x_i))-x_i - End of p.2, the claim "our method can be easily extended to tackle heterogeneous DA ..." seems too strong. By adding one "unknown" class won't solve the issue of having multiple new classes; same to training on "a specified category".

[Author Response · NeurIPS 2019]

We sincerely thank all the reviewers for their insightful comments to help us improve the paper. Here we clarify some
unclear points and will update the paper accordingly in the final version.

**To Reviewer #1.  1. Architectures for generators and discriminators.** We adopt the generator and discriminator
architectures from CycleGAN [38]: 9 residual blocks for generator and 4 convolution layers for discriminator.

**2. $F_A$ and $F$ in Equ. (9).** As explained in Sec. 3.1, $F_A = F$, we will replace $F_A$ with $F$ as suggested.

**To Reviewer #2.  1. Are multiple sources more beneficial?** From the results in Table 2 in the main paper, we can
see that Source-combine domain adaptation (DA) could give worse performance (37.3%) than GTA-only DA (38.7%)
with the same method (CyCADA), which implies that naive combination of different sources is not guaranteed to boost
the target performance. This is largely due to the fact that domain gap also exists among different source domains.
However, since the proposed MADAN can perform domain aggregation to align different sources, it improves the
performance under single-source setting (CyCADA w/ DSC in Table 3, w/o domain aggregation) from 40.0 (GTA) and
31.8 (SYNTHIA) to 41.4 under multi-source setting (MADAN in Table 2) with the same experiment configurations.

**2. Reorganization of Figure 1.** We will reorganize the layout of Figure 1 in the main paper to make it more clear. We
will also explain in detail the meanings of different colors and arrows in the caption and add some legends.

**3. Design of loss functions for different discriminators.** We thank the reviewer for pointing this out. We agree that
using a more sophisticated combination of different discriminators' losses to better aggregate the domains with larger
distances might improve the performance. We leave this as our future work and would explore this direction by dynamic
weighting of the loss terms and incorporating some prior domain knowledge of the sources.

**To Reviewer #3.  1. Feature alignment.** In the feature-level alignment loss function Equ. (8), $F(\mathbf{x})$ is the output of
the last convolution layer in the VGG model, which is a 4096 dimensional feature vector. Whereas, in Equ. (7), $F$ is the
FCN segmentation model, *i.e.* 3 up-sampling and fusing operations following the last convolution layer. We will make
it more clear in the final version.

**2. The computation cost.** We agree that since the proposed framework deals with a harder problem, *i.e.* multi-source
DA, more modules are used to align different sources, which results in a larger model. In our experiments, MADAN is
trained on 4 NVIDIA Tesla P40 GPUs for 40 hours using two source domains which is about twice the training time as
on a single source. However, MADAN does not introduce any additional computation during inference, which is the
biggest concern in real industrial applications, *e.g.* autonomous driving.

**3. On the poorly performing classes.**  There are two main reasons for the poor performance on certain classes: 1)
lack of images containing these classes and 2) structural differences of objects between simulation images and real
images (*e.g.* the trees in simulation images are much taller than those in real images). Generating more images for
different classes and improving the diversity of objects in the simulation environment are two promising directions for
us to explore in future work that may help with these problems.

**4. Ablation study results.**  We agree that it would be ideal to propose a framework that could uniformly improve the
performance on every class. However, semantic segmentation is a challenging pixel-level prediction task, and none
of the existing DA methods can achieve the best performance on every class. Therefore, mIoU is used as the most
important metric. Although some of the classes have a little performance degradation during the progressive addition of
modules in MADAN, the mIoU consistently increases.

**5. Performance on class "sky".**   We observed
that in some images, artifacts are introduced in "sky"
after image translation.  This is probably due to
performing alignment among different sources. As
shown in the right Figure 1 (b)(d), the sky is adapted
with dark colors, making it look like trees. We plan
to address this issue with constraints of intrinsic
spatial layout priors [47], *e.g.* that sky is more likely
to be on the top of an image than ground.

(a)          (b)          (c)          (d)

Figure 1: Examples of bad image translation in "sky": (a) and (c)
are original images; (b) and (d) are adapted images by MADAN.

**6. More adaptation results.**   We conducted more adaptation ex-
periments from GTA, SYNTHIA, and Cityscapes to BDDS. From
the results in the right Table 1, we have similar observations to those
in Section 4.2: non-adaptation methods perform the worst, single-
source adaptation methods (CyCADA w/ DSC) perform better, and
our MADAN performs the best.  More progressive results will be
added in the final version.

Table 1: Domain adaptation results from GTA,
SYNTHIA, and Cityscapes to BDDS.

| Methods | Sources | mIoU |
|---|---|---|
| Source-only (non-adaptation) | GTA | 22.3 |
| | SYNTHIA | 17.1 |
| | GTA+SYNTHIA | 24.6 |
| | GTA+SYNTHIA+Cityscapes | 35.9 |
| Single-source DA (CyCADA w/ DSC) | GTA | 32.3 |
| | SYNTHIA | 27.7 |
| | Cityscapes | 37.8 |
| Mulit-source DA (MADAN) | GTA+SYNTHIA | 39.4 |
| | GTA+SYNTHIA+Cityscapes | 43.2 |

[Meta-Review · NeurIPS 2019]

All reviewers agree that the introduction of the multi-source semantic segmentation setting is worthy. They also praise the dynamic semantic consistency loss, and the adversarial domain aggregation. Overall, this is a solid contribution.